# Approaching national climate targets in China considering the challenge of regional inequality

Biying Yu [1,2,3] ✉, Zihao Zhao[1,2], Yi-Ming Wei [1,3] ✉, Lan-Cui Liu [4] ✉, Qingyu Zhao[1,2], Shuo Xu [1,2], Jia-Ning Kang [1,2] & Hua Liao[1,2,3]

Achievement of national climate targets and the corresponding costs would entirely depend on regional actions within the country. However, because of substantial inequalities and heterogeneities among regions, especially in developing economies, aggressive or uniform actions may exacerbate inequity and induce huge economic losses, which in turn challenges the national climate pledges. Hence, this study extends prior research by proposing economically optimal strategies that can achieve national climate targets and ensure the greatest local and national benefits as well as regional equality. Focusing on the biggest developing country China, we find this strategy can avoid up to 1.54% of cumulative GDP losses for approaching carbon neutrality, and more than 90% of regions would obtain economic gains compared either with existing independently launched targets or with the uniform strategy that all regions achieve peak carbon emissions before 2030. We also provide optimal carbon mitigation pathways to regional peak carbon, carbon intensity and energy consumption.

To avoid huge long-term and irreversible climate risks, nations all over the world have made their climate pledges[1–3]. However, the achievement of national or overall mitigation targets and the corresponding costs would entirely depend on actions of regions within the country[4–8]. Because of substantial social, economic, and technological heterogeneities among regions, especially in developing economies (following the category given by World Economic Outlook released by International Monetary Fund) like China, India, and countries in Africa, the capability and potential of emission reduction in each region are quite unequal, resulting in large gap of carbon abatement costs[9–11]. Existing studies have indicated that achieving an early carbon peak or setting aggressive goals for countries or regions that are economically and technologically not well prepared would hinder the economic growth or even induce great economic losses[11–13]. This would exacerbate imbalances and inequality in regional development for the whole country and in turn challenge the achievement of national climate targets in a fair and economic way[14–16]. Consequently, the interregional collaborative strategies that can best balance the regional equality and economy as well as achieving national cost-effectiveness and climate mitigation targets are urgently required.

Prior literature has proposed a series of climate mitigation strategies at the global or national scale[15,17–21]. Some studies have planned national or regional emission reduction targets by allocating the total emission gap across regions following several equity principles[22–26]. These studies can propose a static fair and effective carbon quota or carbon intensity targets for each region. Nevertheless, they are unable to outline the economically optimal emission reduction pathway for the regions. In contrast to effort-sharing studies, another stream of scholars developed top-down or bottom-up models to derive the carbon emission reduction pathways for multiple regions by considering their socio-economic and technological diversity[10,27,28]. Though they have provided valuable insights for this study, overall

[1]Center for Energy and Environmental Policy Research, Beijing Institute of Technology, Beijing 100081, China. [2]School of Management and Economics, Beijing Institute of Technology, Beijing 100081, China. [3]Beijing Key Lab of Energy Economics and Environmental Management, Beijing 100081, China. [4]School of Business Administration, Beijing Normal University, Beijing 100875, China. ✉e-mail: yubiying_bj@bit.edu.cn; wei@bit.edu.cn; liulancui@163.com

technological cost minimization or market equilibrium is the underlying mechanisms, and these can cover part of the direct economic costs or can evaluate the economic impacts of different pathways but without presenting the economically optimal regional pathways. In other words, these studies are limited in deriving the pathways that can maximize the economic benefits for the whole country and regions and meanwhile consider the regional inequality.

Consequently, this study goes beyond previous research by providing a carbon emissions mitigation strategy to fulfill Chinese national climate pledges considering the national and provincial economies and equity. We first apply the National Energy Technology Model (C³IAM/NET) to explore the pathway for approaching national climate targets (see Methods). To deal with the challenges of regional inequality, a regional maturity index is created to evaluate the difficulty faced by each region to reduce its carbon emissions by comprehensively considering the socio-economic, technological, and resource performance. Constrained by national energy consumption and carbon emissions in achieving carbon mitigation target as well as the regional maturity index, a nonlinear model, the Multi-regional Collaborative Optimization of Emission Pathway (Mr. COEP), is further developed to investigate the carbon emissions and energy consumption pathways for each region (see Methods). On this basis, the important question of how regions collaborate to reach national targets in a cost-effective and fair way is answered. Here we take China, the world's biggest developing country with large regional heterogeneity, as the empirical context. China has put forward the ambitious goal of achieving carbon peak by 2030 and carbon neutrality by 2060. Following this target, about two-thirds of Chinese provinces have independently proposed their own carbon reduction targets until

2023, and almost all of them uniformly claimed that they would reach peak $CO_2$ before 2030. This study would propose regional cost-effective strategies for approaching China's climate targets, which can also provide informative insights for other countries, especially for developing countries with substantial regional inequalities.

## Results

### National pathway for achieving carbon peak and carbon neutrality

The optimal carbon peak and carbon neutrality pathway for China is first put forward by applying the C³IAM/NET model and considering the uncertainties of future product (or service) demand in energy-consuming sectors corresponding to different socio-economic development speeds as well as the carbon sink potential (see Methods, Supplementary Fig. 1 and Table 1 for details). Results show that if the natural carbon sink is 1 billion tons available in 2060[29] and the GDP grows at a medium speed (Supplementary Table 1), China needs to achieve its carbon peak before 2029 and the peak $CO_2$ emissions will be approximately 12.2 billion tons, of which 90.2% is fossil fuel related emissions and the remaining is industrial process emissions (Fig. 1a). The peak consumption of fossil energy will be about 4.8 billion tons of standard coal equivalent (tce). Nationwide, the period of 2025–2035 would be a plateau stage, with the average annual growth rate of carbon emissions being 0.2% before the carbon peak target is achieved in 2029. After that, the rate of $CO_2$ emissions reduction would be between 0.5% and 1.4%, with an average annual decline rate of 1.1% during 2030–2035. Carbon emissions would need to decline rapidly from 2035 to 2050, with an average annual decline rate of 4.1%. The years 2050 to 2060 would be an accelerated emission reduction

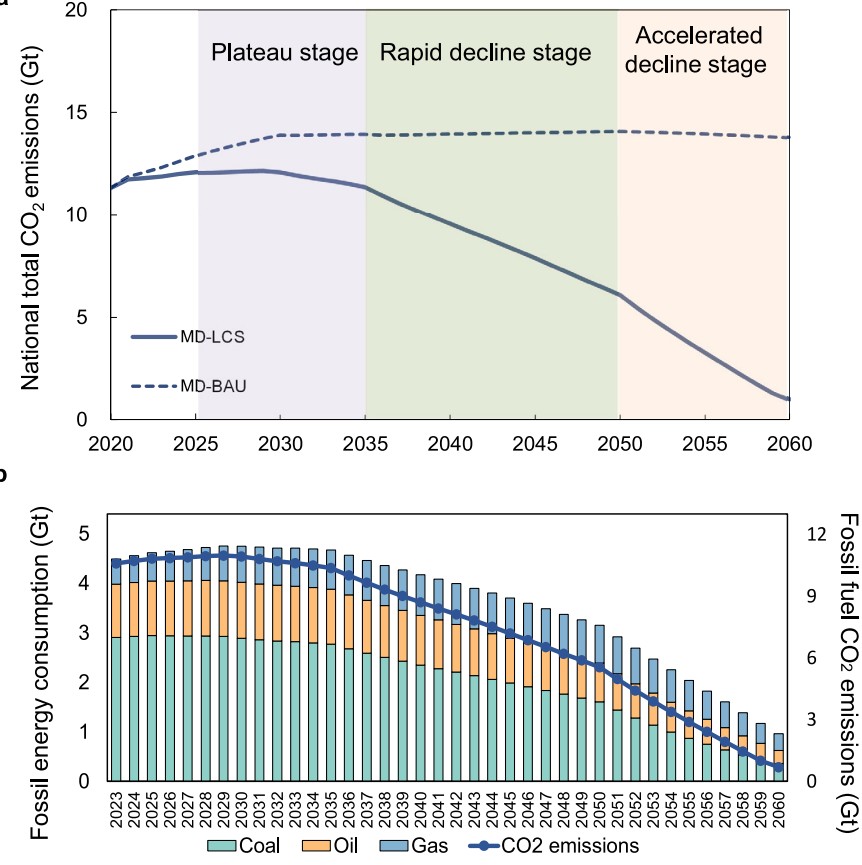

**Fig. 1 | The optimal carbon peak and carbon neutrality pathway for China.**
**a** Pathways for China under BAU and MD-LCS (Medium product or service demand in each sector, and low carbon sink potential in 2060) scenarios. Note that $CO_2$ emissions in this figure is the sum of fossil fuel-related emissions and industrial process emissions. **b** Fossil energy consumption and fossil fuel $CO_2$ emissions under MD-LCS scenario.

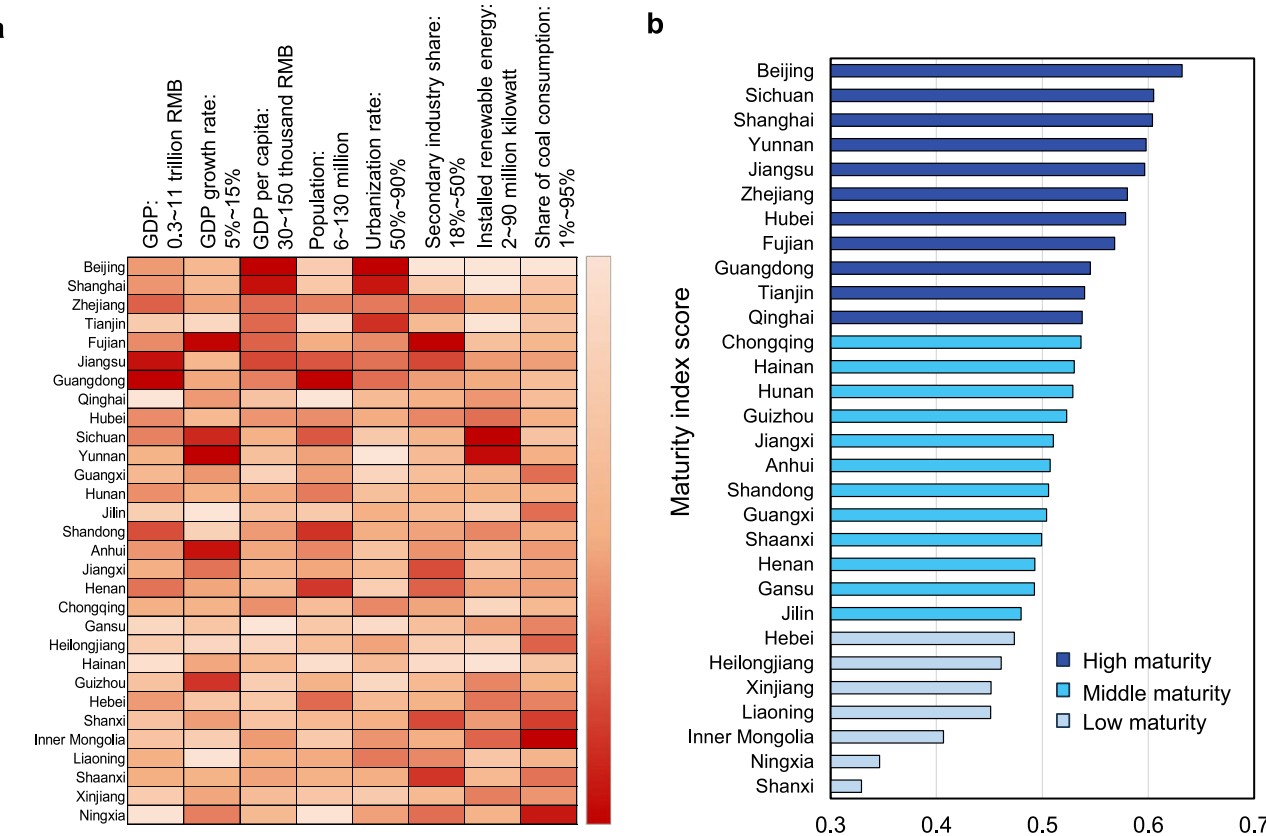

**Fig. 2 | Characteristics and regional maturity index for carbon mitigation.** **a** Differences in socio-economic characteristics, industry structure, energy structure and resource endowments among 30 Chinese provinces. All the data is for year 2020. **b** Carbon peak maturity index scores for 30 Chinese provinces. Provinces are divided into three groups according to its maturity index score: high maturity, middle maturity, and low maturity. The carbon peak year for high-maturity provinces is thought to be earlier than that of mid-maturity provinces; and carbon peak year for middle-maturity provinces is thought to be earlier than that of low-maturity provinces. Note that due to the lack of data for Tibet, Hong Kong, Macao, and Taiwan, this study only considers 30 Chinese provinces excluding these provinces.

period, with an average annual decline rate being greater than 16%. In 2030, the consumption of coal, oil, and natural gas would account for 44.2%, 17.3%, and 11.1% of total energy consumption, respectively; and the non-fossil energy would increase from 15.9% in 2020 to 80.1% in 2060 (Fig. 1b). The energy consumption by fuel type and the fossil fuel related emissions are further used as the constraint for subsequent provincial pathway optimization.

**Regional inequality for carbon mitigation**

To implement the above national pathway, each region needs to make great efforts. However, there are large differences across Chinese provinces in terms of economic development (indicated by GDP related indexes), functional orientation, energy and industrial structure, technology, and other characteristics[30], resulting in substantial regional inequality for carbon mitigation (Fig. 2a). More developed provinces (with both GDP and per capita GDP above the national average) are characterized with a higher share of tertiary industry, lower energy intensity, and advanced technologies, such as Shanghai, Zhejiang, Guangdong, etc.[31]. Underdeveloped provinces (with both GDP and per capita GDP below the national average) usually rely on heavy industry and suffer from outdated technologies and high energy intensity, such as Ningxia, Gansu, Heilongjiang, and Xinjiang[32,33]. Due to these variations, some provinces can achieve carbon peak and carbon neutrality early, while others cannot because their economic development still requires substantial fossil fuel inputs. Hence, it is challenging and unfair for them to achieve the carbon mitigation targets at the same time with other regions.

To reflect such regional inequalities and heterogeneities, a maturity index is proposed by using TOPSIS method for quantifying the capability and potential of each province to achieve the carbon peak and carbon neutral target (see Methods). The lower the provincial maturity index score is, the more challenging for achieving the carbon peak targets. The performance of each province is shown in Fig. 2b and Supplementary Fig. 2. It is found that Beijing, Shanghai, Jiangsu, Zhejiang, Fujian, Guangdong, and Tianjin belong to high-maturity provinces with developed economies and are at the forefront of transitioning to a low-carbon industrial structure. These provinces have low energy intensity, and their economic growth is basically decoupled from carbon emissions. Sichuan, Yunnan, Hubei, and Qinghai also belong to the category of high maturity provinces with high renewable energy resources endowment. Of these, Yunnan, Sichuan, and Hubei have the richest hydropower resources in China. The second category comprises middle-maturity provinces whose growth rate of carbon emissions is smaller than that of the GDP, including Chongqing, Hainan, Hunan, Guizhou, Jiangxi, Anhui, Shandong, Guangxi, Shaanxi, Henan, Gansu, and Jilin. Most of these provinces have started decoupling carbon emissions from economic development, but the industrial and energy transitions need to be accelerated. The remaining provinces are all low-maturity provinces that have not shown obvious decoupling of carbon emissions and GDP. Based on the division of these three groups of provinces, the constraint of carbon peak sequence is set to reflect regional heterogeneity and equality when optimizing the collaborative mitigation strategy.

## Economic impacts of different regional carbon mitigation strategies

Because there are various propositions in the real decision-making process for carbon mitigation strategy, this study designed another three scenarios in addition to the Collaborative optimization (COP) scenario that can accommodate regional inequality and maximize local and national economic benefits. The first scenario is that All provinces will uniformly achieve the carbon peak target before 2030 (AP30), which is a widely discussed policy strategy in the reality. The second scenario is Following the current independently set peak target (FCT), in which 21 provinces will follow their proposed targets on carbon peak timing and energy consumption (see Supplementary Table 2). For those provinces that have not yet decided their climate target, they are unconstrained. The third scenario is that Only energy-intensive provinces are required to achieve carbon peak before 2030 (EIP30). The main energy consuming provinces in China (including Hebei, Shanxi, Inner Mongolia, Liaoning, Jiangsu, Zhejiang, Anhui, Shandong, Henan, Guangdong, Shaanxi, and Xinjiang) accounted for approximately 70% of the country's total carbon emissions in 2020. It is argued that if these provinces can reach their carbon emissions peak on time, then the entire country can achieve the target on time. Hence, we set up the EIP30 scenario. Subsequently, Mr. COEP model is used to optimize regional emission pathway under the constraint of national energy consumption and fossil fuel related $CO_2$ emissions as well as the itemized objectives set in the four scenarios.

Results show that under different scenarios, there are substantial differences in terms of the carbon peak year (Fig. 3a). Though these four strategies can all realize China's climate target with a maximum GDP, the resulting economic benefits or losses for the whole country as well as for each province are totally different (Fig. 3b–d). The strategy of all provinces uniformly achieving a carbon peak before 2030 (AP30) will result in a total GDP of approximately 10963 trillion RMB (constant price in 2020) for 30 provinces of China during 2023–2060. If following the current carbon peak and energy target that some provinces have independently proposed (FCT), the cumulative GDP would be 0.63% higher than that of the AP30 scenario. While for the strategy of only energy-intensive provinces being required to peak their carbon emissions before 2030 (EIP30), the cumulative GDP would be 0.91% higher than that of AP30 scenario. The economic benefits of the collaborative strategy (COP) are the highest, which is 1.54% higher. In other words, the collaborative strategy that considers provincial equity on carbon mitigation based on their socio-economic and technological characteristics, can avoid up to 1.54% of GDP loss until 2060 and bring the greatest overall economic benefits on the way to reaching China's carbon peak and carbon neutrality target.

The economic impacts on provinces are revealed to be very uneven (Fig. 3c, d). Among all scenarios, provincial economic benefits under the COP scenario are generally larger (Fig. 3c). Specifically, compared with the AP30 scenario, 27 of 30 provinces will have additional GDP gains, ranging from 0.34 to 20.87 trillion RMB and 6.28 trillion RMB on average during 2023–2060 if following the collaborative strategy. To ensure the national economically optimal and climate targets, only Sichuan, Hubei, and Jilin would suffer economic losses, which are 0.67, 0.29, and 0.06 trillion RMB respectively, accounting for 0.05–0.12% of the provincial cumulative GDP under COP scenario. Provinces such as Guangdong, Jiangsu, Zhejiang and Shanghai could have much larger cumulative economic gains, which may increase by up to 20.9, 18.9, 11.7, and 11.6 trillion RMB, respectively. Compared with the COP scenario that considered the provincial equity, the cumulative GDP corresponding to the FCT scenario (following the current independently proposed targets) would decrease by 101.2 trillion RMB between 2023 and 2060. Therein, 27 provinces would experience economic losses with the average cumulative GDP loss being 3.75 trillion RMB. Only three provinces, including Guangxi, Gansu, and Xinjiang, are likely to generate a small amount of economic

benefits, ranging between 0.1–0.5% of the provincial cumulative GDP. These findings once again demonstrate that setting an early carbon peak target for each province or having provinces operate separately would cause national losses and harm the local economy for most provinces.

## Regional cost-effective pathways of carbon emissions and energy consumption

Because the collaborative and equitable carbon emission pathway can achieve the most favorable economic outcome, this section introduces the carbon peak year, carbon intensity, and energy structure for each province under this strategy (Fig. 4). To ensure the maximum benefits for national and regional development, all the provinces are encouraged to follow a sequential carbon peak action timeline.

Some pioneer provinces could reach their carbon peak earlier, by 2027, including Shanghai, Zhejiang, Tianjin, Fujian, Jiangsu, Guangdong, Qinghai, Hubei, Sichuan, and Yunnan. These provinces have developed economies, better renewable endowments, or a low share of coal. The five-year reduction rate of $CO_2$ emission intensity for provinces in this category would gradually increase from 21% during 2020–2025 to 34% during 2035–2040. Among them, Beijing, Zhejiang, and Shanghai are the provinces with higher reduction rates, while Qinghai and Tianjin have lower reduction rates. Due to the occurrence of a platform period for carbon emission and the possibility of a slowdown in GDP growth, the reduction rate of carbon intensity during 2030-2035 for the pioneer provinces would decrease by 0.4% compared to the reduction rate during 2025–2030.

Provinces including Jilin, Hunan, Guangxi, Anhui, Jiangxi, Shandong, Henan, Chongqing, Hainan, Shaanxi, Guizhou, and Gansu can keep pace with the whole country to set their carbon peak date. Most of these provinces have obvious economic growth momentum and a slightly higher share of coal. Some of them are in or adjacent to regions with considerable renewable resources, such as Hunan, Chongqing, and Jiangxi. These provinces need to gradually increase their five-year reduction rate of $CO_2$ emission intensity from 17.7% during 2020–2025 to 31.8% during 2035–2040. Among them, Guizhou, Jiangxi, and Chongqing are provinces with higher reduction rates, while Shandong and Jilin have lower reduction rates.

Some vulnerable provinces with less-developed economies, heavy industrial structures, and a high share of coal consumption are allowed to peak later than 2030 to have more time for smooth transition, including Hebei, Shanxi, Inner Mongolia, Liaoning, Heilongjiang, Ningxia and Xinjiang. However, their carbon peak date should be no later than 2034. These provinces have relatively low five-year reduction rates of $CO_2$ emission intensity. However, the rates need to increase substantially over time. Compared with the five-year reduction rates during 2020–2025 (9.0%), 2025–2030 (11.8%), and 2030–2035 (15.4%), the $CO_2$ emission intensity for these vulnerable provinces needs to decrease more rapidly after 2035, which should increase to 24.8% during 2035–2040 on average.

To realize the above carbon intensity reduction, adjustment of the energy structure is a crucial task. We further estimate the energy consumption and energy structure for each province (Figs. 5, 6 and Supplementary Fig. 3). For provinces with a high share of renewable energy (e.g., Qinghai, Yunnan, and Sichuan), their share of non-fossil energy needs to reach 78%, 73%, and 67%, respectively, by 2040. Zhejiang, Anhui, and Xinjiang are supposed to accelerate the promotion of non-fossil energy, among which, Zhejiang needs to increase its non-fossil energy from 28% in 2025 to more than 55% in 2040, and Anhui needs to increase from 13% in 2025 to more than 28% in 2040. Though all provinces are required to increase non-fossil energy, China is still a coal-dominant country and will remain so even until 2040 due to energy security considerations. For some provinces that are energy suppliers to other regions, such as Shanxi, Inner Mongolia, and

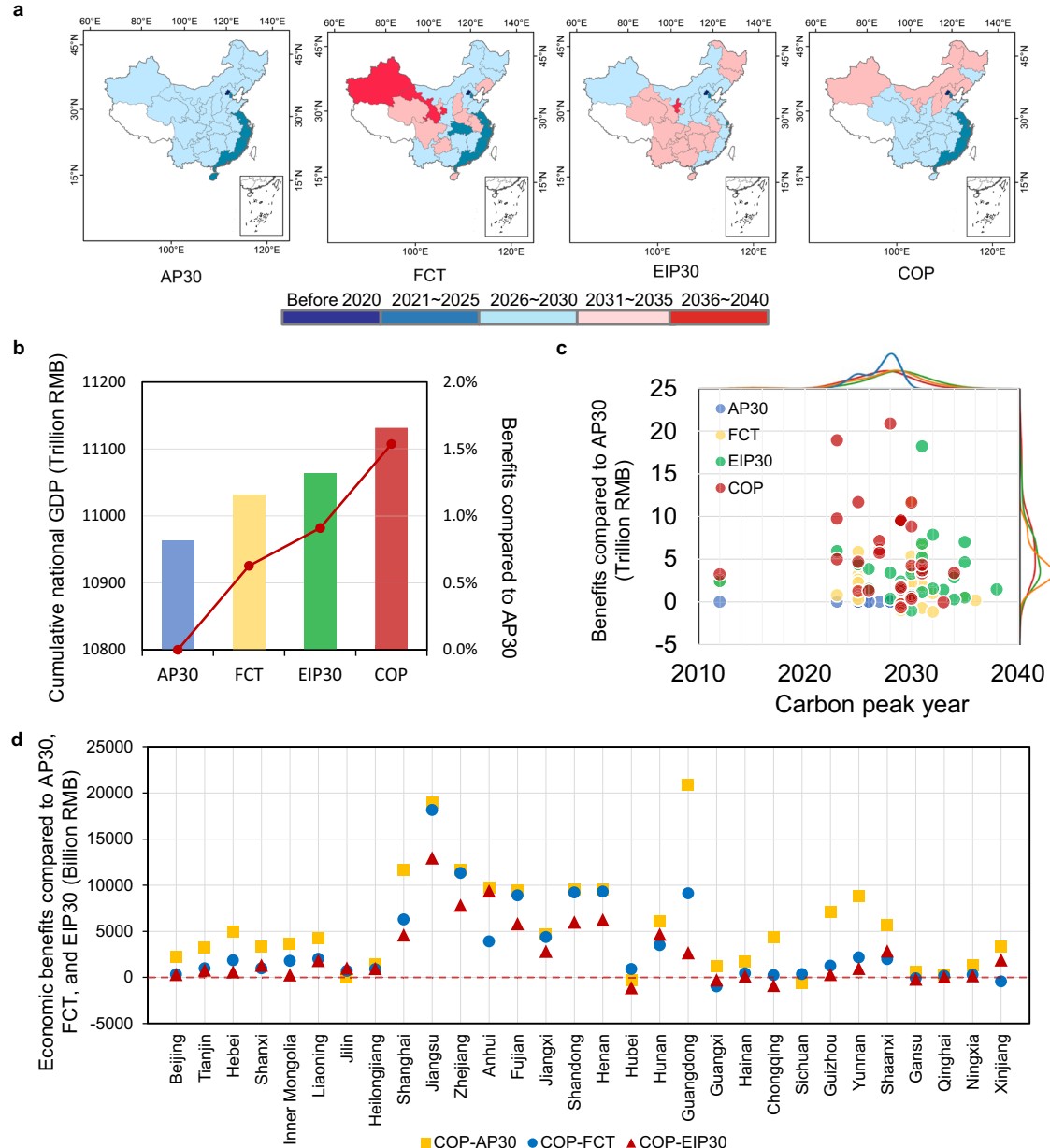

**Fig. 3 | Economic impacts and carbon peak time for each province under different scenarios. a** Carbon peak time for each province under four scenarios. **b** Cumulative national GDP from 2023 to 2060. The bar chart shows the total GDP, and the line shows the percentage improvement in economic benefits under the FCT, EIP30, and COP scenarios compared to the AP30 scenario. **c** Distribution of provincial economic benefits (2023-2060) compared to that of AP30 and carbon peak years. The dots in the figure represent values of the provinces under the four scenarios. The curve on the secondary axis of the vertical coordinate represents the distribution of benefits under the three scenarios compared with the AP30 scenario. The curve on the sub-axis of the horizontal coordinate represents the distribution of the peak year for each province under different scenarios. **d**, Provincial economic benefits (2023-2060) of the COP scenario compared with that of the AP30, FCT, and EIP30 scenarios.

Shaanxi, their share of coal consumption cannot be less than 80%, 71%, and 57%, respectively, by 2030.

## Discussion

Feasible and cost-effective mitigation actions from each region are the key to achieve the national climate targets. Ignoring the heterogeneity on socio-economy, technology, and resource endowment among regions, and setting aggressive regional carbon reduction targets or uniform targets would amplify the inequality and induce economic losses for regions and the whole country, which is especially severe for developing countries. To deal with such challenges from regional inequality for approaching national climate targets, this study

contributes to investigate the optimal carbon mitigation strategies that can best balance both the regional and national cost-effectiveness as well as the regional inequality when fulfilling national climate pledges. China, the biggest developing country that have displayed large regional variations, is taken as the case study. We find that China is likely to peak the carbon emissions before 2029 at no more than 12.2 billion tons of $CO_2$ (including industrial process emissions). Following the proposed cost-effective strategy, 90% of Chinese provinces can obtain economic benefits compared with the strategy that all provinces uniformly peak their carbon emissions by 2030 (6.28 trillion RMB gains on average) or compared with the current targets proposed by each province independently (3.75 trillion RMB gains on average) or

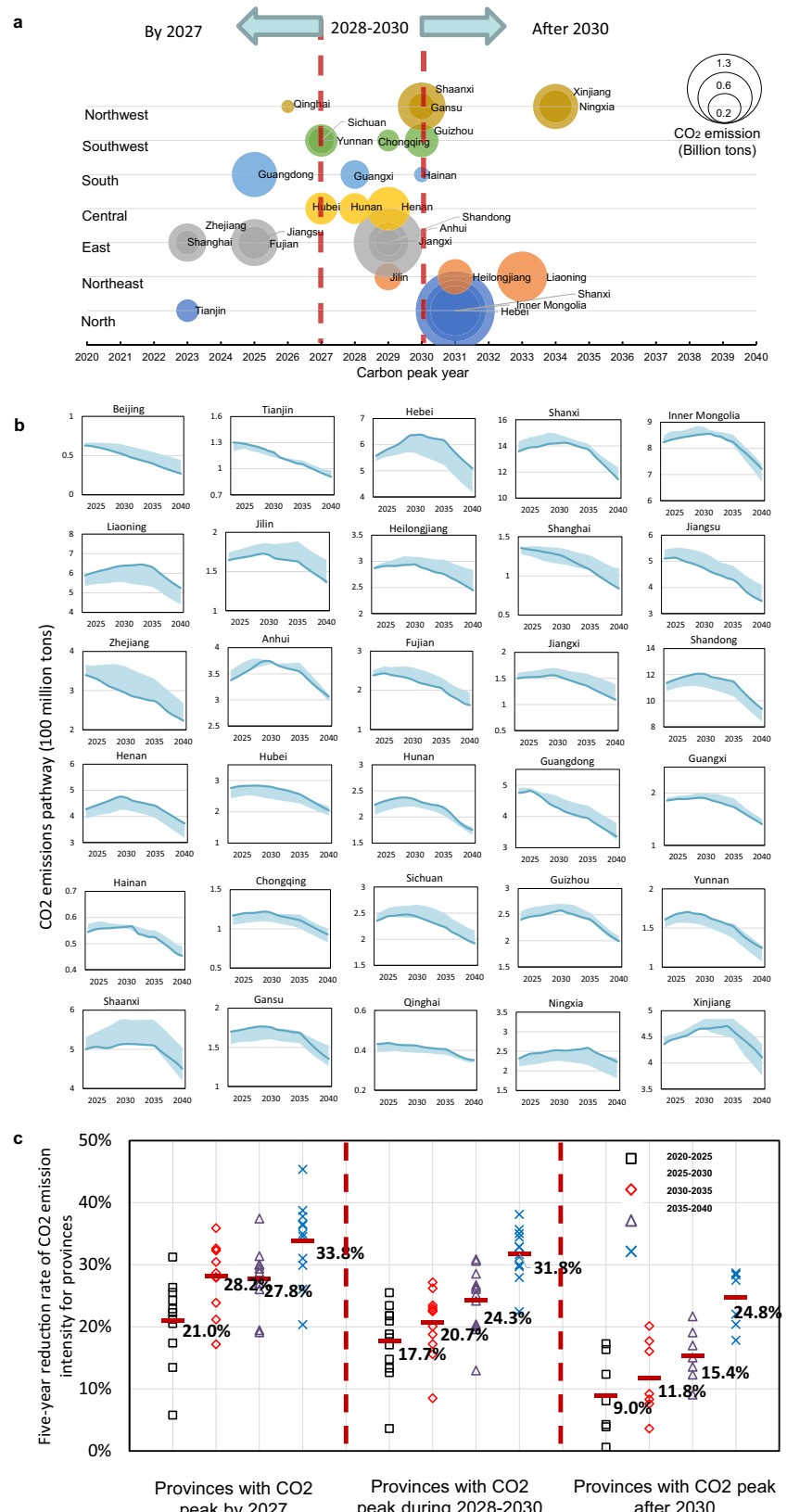

**Fig. 4 | Collaborative carbon mitigation strategy for each province. a** Carbon peak year and CO₂ emissions at peak year by province. The size of the bubble denotes the provincial CO₂ emissions at the peak year. The horizontal coordinate of the bubble indicates the year in which the carbon peak target will be achieved. **b** Carbon emission pathways. The shading indicates the range of emission pathways for the other three scenarios. **c** Five-year reduction rate of CO₂ emission intensity for provinces in three peak year groups. The icons indicate the values for each province during different time periods. The red line indicates the average value of that group of provinces during the specific time period.

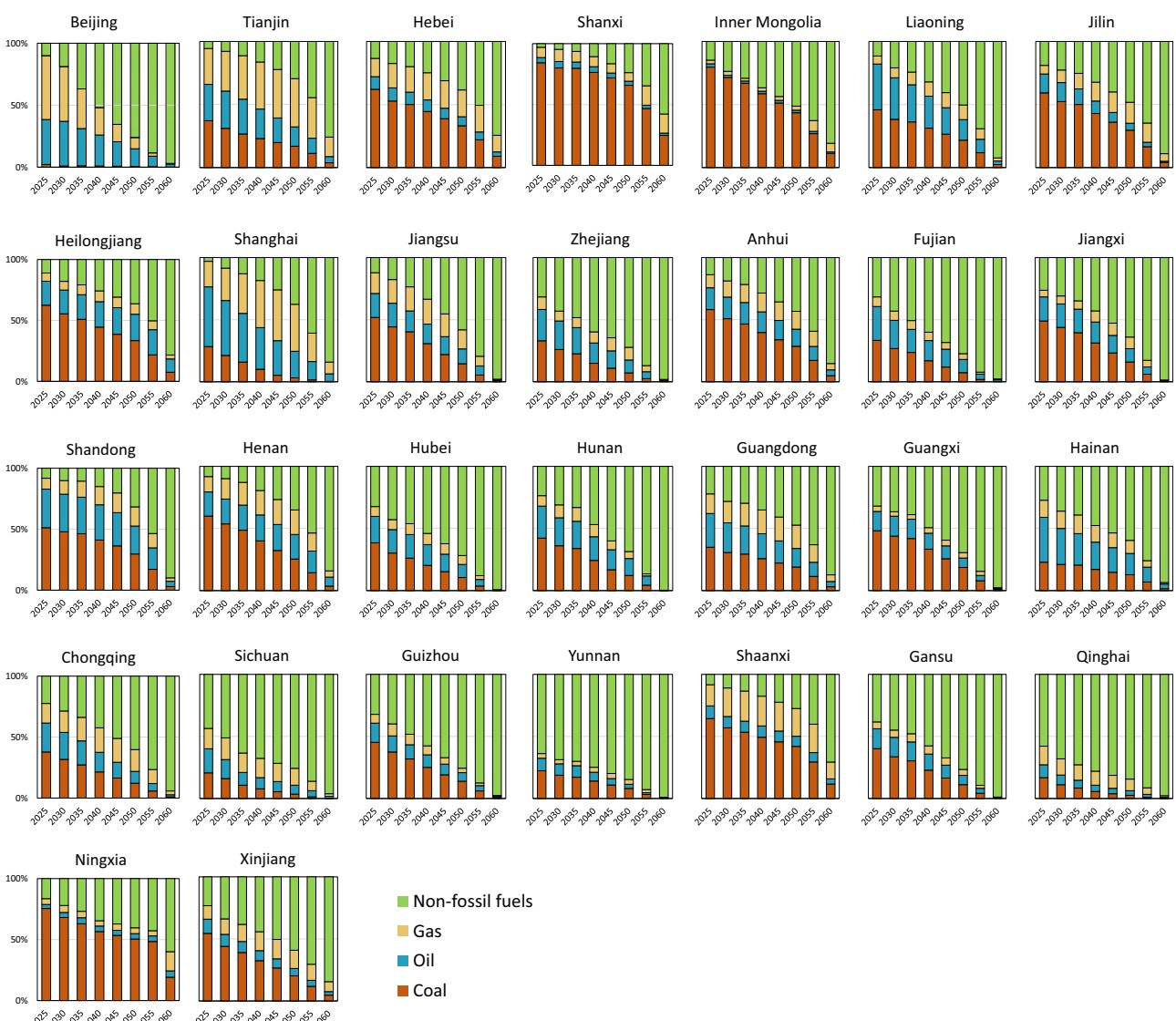

**Fig. 5 | National and provincial energy structure.** Each sub-figure shows the energy consumption structure of each province during 2025–2060 obtained from Mr. COEP model, including coal, oil, natural gas, and non-fossil energy.

compared with the strategy that energy-intensive provinces peak before 2030 (3.11 trillion RMB gains on average). In total, approximately 1.54% of cumulative GDP losses during 2023-2060 can be avoided for the whole of China. Therefore, it is highlighted that the collaboration among Chinese provinces is quite important for ensuring better development for each region and the whole country. In addition to the above strategies, a scenario without any carbon peak time constraints is analyzed (see Supplementary Figs. 4, 5 for details). The results show that though the total economic benefits would increase compared to that of COP scenario, some provinces with less-developed economy and fewer renewable resources (e.g., Shanxi and Hebei), which indicate lower carbon peak maturity, need to peak their carbon emissions earlier than that of COP, while some provinces with developed economy and rich renewable energy (e.g., Sichuan and Hubei) could delay their carbon peak year. This may be less feasible in practice. Therefore, it is once again highlighted that the carbon peak maturity that can represent the regional inequality needs to be considered during the decision making for the provincial carbon mitigation strategies.

Considering that 21 Chinese provinces have already announced their carbon peak or energy transition targets independently, we further put forward the corresponding adjustment suggestions to increase provincial GDP and equity on the way to national carbon neutrality (Fig. 6a). Specifically, a total of 10 provinces could slightly bring forward their current proposed carbon peak targets. For example, Qinghai, Guangxi, Hunan, and Henan could refine their targets to peak the carbon emissions by 2026, 2028, 2028, and 2029, respectively, which is 1–4 years ahead of the existing timetable. In addition, Hebei, Ningxia, Hainan, Inner Mongolia, Shandong, Heilongjiang and Liaoning could postpone their carbon peak time by 1–5 years to avoid potential economic losses caused by premature carbon peak actions but should peak no later than 2034. For the provinces that have no clear targets so far, to realize the largest economic benefits for China, Xinjiang could achieve a carbon peak after 2030, but not later than 2034; Sichuan, Yunnan, Fujian, Jiangsu, and Hubei need to achieve the carbon peak earlier, by 2027; and Anhui and Gansu could achieve the carbon peak target by 2030.

Regarding some provinces that have set clear goals for the share of fossil fuel or non-fossil energy, they could contribute more to national economic development by shifting their actions (Fig. 6b). It is recommended that Hunan, and Guangxi adjust their current targets of reaching 25% and 35% of non-fossil energy by 2030 to 27% and 37%. Ningxia could raise its current target of 15% and 20% of non-fossil energy in 2025 and 2030 to 16% and 22%, respectively. Qinghai and

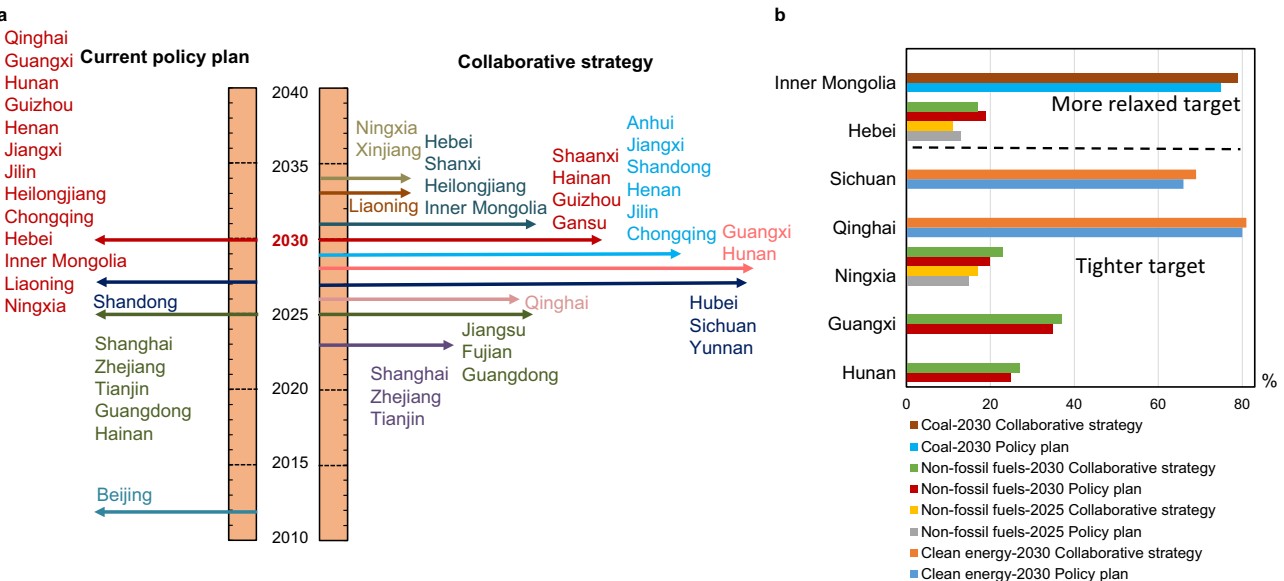

**Fig. 6 | Suggested settings of carbon mitigation and energy transition targets for provinces. a** Suggestions for the carbon peak year. **b** Suggestions for the energy transition targets.

Sichuan could raise its current target of reaching 80% and 66% of clean energy by 2030 to 81% and 69%, respectively. On the other hand, it is recommended that Hebei and Inner Mongolia relax their targets appropriately. For example, Hebei could lower its current target of non-fossil energy consumption (over 13% by 2025 and over 19% by 2030) to 12% and 17%. Inner Mongolia could slightly relax its target for the share of coal consumption, changing from 75% to 79% by 2025. Some provinces such as Jilin and Liaoning, could continue their current set targets with no need for adjustment.

Though the proposed regional collaborative strategy is the most cost-effective solution for China to reach carbon neutrality and can enhance economic benefits for most provinces, there remain three provinces that would suffer economic losses, including Jilin, Sichuan, and Hubei. Their economic losses would range from 0.05% to 0.12% of provincial cumulative GDP during 2023–2060, indicating a small proportion. However, to achieve a national win-win situation and satisfy all concerned, we suggest that China establishes a carbon compensation mechanism and a collaborative carbon mitigation fund among vulnerable and high-yield provinces (e.g., Guangdong, Jiangsu, Shanghai, Zhejiang, and Anhui). High-yield and developed provinces need to provide technical, financial, and professional support to those provinces that would suffer losses.

The results of this study are targeting China, however, the research scheme and regional strategies can be applied to other countries, especially to developing countries with substantial regional inequalities. Besides, this research also has some limitations that can be improved in future study. For instance, only energy-related carbon emissions were considered here for provincial emission pathway optimization; and the industrial process emissions across different regions need to be further investigated. In addition, the accelerated process of carbon neutrality may lead to disruptive industry transfer across provinces due to uneven distribution of low-carbon technologies and renewable resource, thereby changing the economic linkages among provinces. Though this study has taken into account the influence of dynamic socio-economic development in future following the literature prediction, further uncertainties could be investigated in the next step.

## Methods
To obtain the best regional strategy to achieve national carbon neutral targets, we first apply the C³IAM/NET model to propose an optimal

emissions pathway for achieving China's carbon peak and carbon neutrality targets. Then, we consider the multidimensional hetero-geneity of social, economic, and technological aspects of each region to assess the difficulty and maturity of carbon mitigation. Next, we investigate a nonlinear relationship between socio-economic devel-opment, technology, and energy for each region. We then construct a multi-regional collaborative emission reduction pathway optimization model (Mr. COEP) by incorporating the outputs from the previous three steps (including the national carbon emissions, energy con-sumption, carbon peak maturity, and the nonlinear relationship) as constraints, and finally propose a cost-effective strategy that can bal-ance the overall and local economy and regional equity.

### C³IAM/NET model for optimizing the national carbon peak and carbon neutrality pathway
C³IAM/NET (China's Climate Change Integrated Assessment Model/National Energy Technology) model is a sub-module of C³IAM, which was developed by the Center for Energy and Environmental Policy of Beijing Institute of Technology (CEEP-BIT). The model is a bottom-up energy technology optimization model[34–38] (see Supplementary Fig. 6 for the model framework). The principle of the C³IAM/NET model is to optimize the technology portfolio for the entire energy system by minimizing the total cost, including sectors such as primary energy supply, electricity[39,40], heating[37], iron and steel[41], cement[42], nonferrous[43], chemical[44], residential[45], transportation[46,47], and other industries. A total of 817 technologies are involved. Each end-use energy-consuming sector is independent of each other. Production in energy supply sectors (e.g., primary energy supply, electricity and heating industries) is endogenously driven by the energy demand of end-use sectors. The C³IAM/NET model first predicts the future demand for products and energy services in each end-use sector based on comprehensive consideration of changes in socioeconomic pat-terns such as economic development, industrial upgrading, acceler-ated urbanization, and popularization of E-life; and secondly simulates the energy and material flows of various technologies in the produc-tion processes or consumption processes of each end-use sector; and thirdly introduces changing trends and policy orientation such as technological upgrading, fuel substitution, and cost reduction. The C³IAM/NET model realizes the joint optimization of all technologies in the supply and demand sectors[34].

## Mr. COEP model for obtaining the regional pathway

Constrained by the carbon neutral pathway and energy consumption derived from the C³IAM/NET model, we further develop the Multi-regional Collaborative Optimization of Emission Pathway (Mr. COEP) model to explore the regional pathway under different scenarios (see Supplementary Fig. 7 for the model framework). Specifically, we first create a maturity index to evaluate the capability and potential for regions to mitigate their carbon emissions. The carbon peak sequence based on the maturity index is incorporated into the Mr. COEP model to represent the regional inequality related to the socio-economic, technological and resource heterogeneities among regions under the COP scenario. Second, the complex nonlinear relationship between regional energy consumption and key influencing factors is investigated and included into the Mr. COEP model based on the extended Kuznets curve theory. Finally, an optimization model is developed with the objective of maximizing the overall national GDP and ensuring the acceptable impact on regional GDP. Note that in the Mr. COEP model, the economic linkage among different regions is represented through the constraints of national energy consumption and carbon emissions as well as the objective function. Specifically, regional carbon emissions are optimized under the constraints of annual national emissions and energy consumption by fuel type (i.e., coal, oil, gas, non-fossil consumption). Because regional GDP is linked with its energy consumption following the nonlinear relationship, thus, economic changes in one region will lead to changes in its energy consumption by fuel type, which will then change the energy consumption for other regions. This change will further influence the GDP of other regions and then the national GDP. The optimization will iterate the above process until the objective (maximize the national GDP) is satisfied. Finally, regional carbon emissions are calculated based on their fossil energy consumption.

In the Mr. COEP model, decision variables include $Energy_{i,k,t}$ (the consumption of coal, oil, gas, and non-fossil fuels in region $i$ at year $t$) and $GDP_{i,t}$ (GDP of region $i$ in year $t$). The remaining variables are exogenous variables, such as the national total amount of energy consumption by fuel type (which is derived from C³IAM/NET model), population size, urbanization rate, industrial structure, etc. The specific settings of exogenous variables are shown in the Supplementary Tables 1–5. The mathematical descriptions are as follows.

The Mr. COEP model aims to maximize the total GDP of all regions (i.e., the whole country) in the year under study. The objective function is shown in Eq. (1). $GDP_{i,t}$ denotes the GDP of region $i$ in year $t$.

$$\max \sum_{i,t} GDP_{i,t} \tag{1}$$

### Regional carbon peak maturity constraint

To reflect the differences and ensure equity among regions, this study innovatively proposes the regional maturity index and establishes an evaluation system with multiple indicators, including per capita GDP, energy intensity, coal consumption proportion, and installed renewable energy capacity, thus reflecting the level of regional economic development, energy and industrial structures, and technical level. Provinces are then divided into high-maturity, middle-maturity and low-maturity groups based on their maturity index score. Subsequently, we set the constraints of the carbon peak sequence for these three groups of provinces in the model under the COP scenario. The carbon peak year for high-maturity provinces is constrained to be earlier than that of middle-maturity provinces; and carbon peak year for middle-maturity provinces is constrained to be earlier than that of low-maturity provinces. Then the exact carbon peak year for each province is optimized in the model by considering carbon peak sequence constraint as well as national and regional economic benefits. For the other three scenarios, this constraint is invalid.

The TOPSIS (Technique for Order Preference by Similarity to an Ideal Solution) method is adopted to evaluate the maturity index of each province. The calculation steps of this method are as follows:

1. Perform the same trend processing on a set of data with $m$ samples and $n$ indicators, and then perform dimensionless data processing according to the Eq. (2) to obtain the dimensionless decision matrix $\boldsymbol{Z} = (z_{ij})_{m*n}$:

$$z_{ij} = \frac{x_{ij}}{\sqrt{\sum_{i=1}^{m} x_{ij}}} \tag{2}$$

2. Determine the optimal solution $z_j^+$ and worst solution $z_j^-$ for each indicator:

$$\begin{cases} z_j^+ = \max\left\{z_{1j}, z_{2j}, \ldots, z_{mj}\right\} \\ z_j^- = \min\left\{z_{1j}, z_{2j}, \ldots, z_{mj}\right\} \end{cases} \tag{3}$$

3. Determine the weighted Euclidean distances $D_i^+$ and $D_i^-$ between each evaluation object and the optimal and worst solutions:

$$\begin{cases} D_i^+ = \sqrt{\sum_{j=1}^{n}\left[w_j\left(z_{ij} - z_j^+\right)\right]^2} \\ D_i^- = \sqrt{\sum_{j=1}^{n}\left[w_j\left(z_{ij} - z_j^-\right)\right]^2} \end{cases} \tag{4}$$

where $w_j$ is the weight of indicator $j$.

4. Determine proximity $C_i$:

$$C_i = \frac{D_i^-}{D_i^+ + D_i^-} \tag{5}$$

The closer the $C_i$ value is to 1, the closer the province is to the best equity indicator, that is, the object is relatively better. Then, we ranked the maturity index of each province in descending order. After sorting the provinces, we divided them into three groups from high to low according to the maturity index score.

### Constraint on regional economic development

To reflect the economic development targets set by some regions and ensure regional economic growth, we set constraints on GDP growth (see Eq. (6)). Here, $lower\_GDP_{i,t}$ and $upper\_GDP_{i,t}$ respectively represent the lower and upper limits to regional GDP growth. $i$ and $t$ denote the region and year considered. See Supplementary Table 3 for specific settings.

$$lower\_GDP_{i,t} \leq GDP_{i,t+1}/GDP_{i,t} \leq upper\_GDP_{i,t} \tag{6}$$

### Constraint on energy consumption

In each year, the sum of all types of energy consumption in each region should be equal to the total consumption following the national carbon peak and carbon neutrality pathway (see Eq. (7)). Here, $Annual\_consumption_{k,t}(k=1,2,3,4)$ denotes the cumulative consumption of coal, oil, gas, and non-fossil fuels in all regions at year $t$. $Energy_{i,k,t}(k=1,2,3,4)$ denotes the consumption of coal, oil, gas, and non-fossil fuels in region $i$ at year $t$.

$$\sum_i Energy_{i,k,t} = Annual\_consumption_{k,t} \tag{7}$$

In addition to the total energy constraints, some regions have set targets for decreasing the proportion of coal consumption and

increasing the proportion of non-fossil energy consumption in the FCT scenario. Therefore, the constraints on the share of different types of energy consumption in different regions in specific years are considered (see Eq. (8)). $lower\_Energy_{i,k,t}$ and $upper\_Energy_{i,k,t}$ denote the lower and upper proportion of each type of energy in total consumption in year $t$.

$$lower\_Energy_{i,k,t} \leq Energy_{i,k,t} / \sum_k Energy_{i,k,t} \leq upper\_Energy_{i,k,t} \quad (8)$$

### Constraints on fossil fuel-related carbon emissions and carbon intensity

The sum of carbon emissions of each region every year should equal to that of the whole country following the carbon peak and carbon neutrality pathway. The calculation method of $CO_2$ emissions generated by energy activities in this study is shown in Eq. (9). $\beta_k$ denotes the energy factor of energy $k$. The total $CO_2$ emission constraint is shown in Eq. (10), where $Emisson\_Total_t$ represents the national $CO_2$ emissions generated by fossil fuel combustion at year $t$, and $Emisson_{i,k,t}$ denotes the emissions of energy $k$ in region $i$ at $t$ year. Note that the industrial process emissions are excluded in the regional analysis.

$$Emission_{i,k,t} = Energy_{i,k,t}\,\beta_k\,(k=1,2,3) \quad (9)$$

$$Emisson\_Total_t = \sum_{i,k} Energy_{i,k,t} \cdot \beta_k\,(k=1,2,3) \quad (10)$$

For the regions that have clear settings for carbon peak time in the FCT and EIP30 scenarios, the carbon emissions in the peak year will be greater than those in any other year, as shown in Eq. (11). $Emission_{i,k,t^*}$ denotes the emissions of energy $k$ in region $i$ at the carbon peak year $t^*$.

$$\sum_k Emission_{i,k,t} \leq \sum_k Emission_{i,k,t^*} \quad (11)$$

Similarly, the $CO_2$ emissions per unit GDP (i.e., $CO_2$ emission intensity) should not exceed the limit values that have been set in different scenarios. Constraints on $CO_2$ emission intensity are set following Eqs. (12) and (13). $lower\_intensity_{i,t^*}$ and $upper\_intensity_{i,t^*}$ respectively represent the lower and upper limits of the change rate of $CO_2$ emission intensity in target year $t^*$ for region $i$. $GDP_{i,t^*}$ represents the GDP of region $i$ in target year $t^*$. $lower\_intensity_{t^*}^{Country}$ and $lower\_intensity_{t^*}^{Country}$ respectively represent the lower and upper limits of the reduction rate of carbon intensity in target year $t^*$ of the country.

$$lower\_intensity_{i,t^*} \leq \left( \frac{\sum_k Emission_{i,k,t^*}}{GDP_{i,t^*}} - \frac{\sum_k Emission_{i,k,t}}{GDP_{i,t}} \right) \Big/ \frac{\sum_k Emission_{i,k,t^*}}{GDP_{i,t^*}} \leq upper\_intensity_{i,t^*} \quad (12)$$

$$lower\_intensity_{t^*}^{Country} \leq \left( \frac{\sum_{i,k} Emission_{i,k,t^*}}{\sum_i GDP_{i,t^*}} - \frac{\sum_{i,k} Emission_{i,k,t}}{\sum_i GDP_{i,t}} \right) \Big/ \frac{\sum_{i,k} Emission_{i,k,t^*}}{\sum_i GDP_{i,t^*}} \leq upper\_intensity_{t^*}^{Country} \quad (13)$$

### Monotony of carbon emissions

In the carbon emissions pathway, the peak year can be regarded as the turning point, and the ideal carbon peak path should be a curve of first increasing and then decreasing. We set Eq. (14) as the indicator

equation and define the constraint in Eq. (15), where $M$ is an artificial variable and is set as a maximum constant.

$$z_i(t) = \begin{cases} 0, \text{The carbon peak target in region } i \text{ is not achieved in year } t \\ 1 \text{ Region } i \text{ achieved carbon peak target in year } t \end{cases}$$
$$(14)$$

$$\left( \sum_k Emission_{i,k,t-1} - \sum_k Emission_{i,k,t} \right) \bullet \left( \sum_k Emission_{i,k,t} - \sum_k Emission_{i,k,t+1} \right)$$
$$\geq -M \bullet z_i(t) \quad (15)$$

We set $\boldsymbol{Z}_{i^s} = [z_{i^s}(1), \cdots, z_{i^s}(t), \cdots, z_{i^s}(T)]$. $s(s=1,\ldots,S)$ denotes the maturity category. $S$ denotes the total number of categories, that is, $i^s$ represents the region in category $s$. Given the directional quantity $\boldsymbol{T} = [1,\ldots,t,\ldots,T]'$, the constraint is set by Eq. (16) for the year of carbon peak in various regions obtained from TOPSIS results. This constraint further restricts the sequence of carbon peak time following the regional maturity index. Regions belonging to the $s$-1 category will achieve carbon peaking earlier than those belonging to the $s$ category.

$$\boldsymbol{Z}_{i^{s-1}}\boldsymbol{T} < \boldsymbol{Z}_{i^s}\boldsymbol{T}, (1 < s \leq 3) \quad (16)$$

**Nonlinear relationship between socio-economic development, technology, and energy.** There is a certain correlation between regional energy growth and its driving factors. The identified relationship is shown in Eq. (17). $PV_{i,t}, UR_{i,t}, SI_{i,t}, EI_{i,t}, CP_{i,t}, RI_{i,t}$, respectively represent the population, urbanization rate, secondary industry share, energy intensity, proportion of coal consumption, and installed capacity of renewable energy in region $i$ in year $t$. $a_i, b_i, c_i, d_i$ denote the coefficients of constant terms and parameters in the regression equation. Considering data availability for the future trend, variables $PV_{i,t}, UR_{i,t}, SI_{i,t}$ are selected to be included in the regression model, with coefficients of $e_i, f_i, g_i$ respectively, and other indicators reflecting regional heterogeneity ($EI_{i,t}, CP_{i,t}, RI_{i,t}$) are introduced into the model through the carbon peak maturity index.

$$\sum_{i,k} Energy_{i,k,t} = a_i + b_i GDP_{i,t} + c_i GDP_{i,t}^2 + d_i GDP_{i,t}^3$$
$$+ f\left(PV_{i,t}, UR_{i,t}, SI_{i,t}, EI_{i,t}, CP_{i,t}, RI_{i,t}\right) \quad (17)$$

This study extends the Kuznets curve by considering the possible effects of the secondary industry share, population size, and urbanization, thus obtaining a regression model that is more appropriate to the actual situation[48,49]. We calculate the extended Kuznets curve for each region. The statistical tests were two-sided and the results of the nonlinear relationship between variables are shown in Supplementary Table 6. The quantified relationship is included in the Mr. COEP model to optimize the regional energy consumption. To represent the future trends of social and economic development, the dynamic changes of secondary industry share, population, and urbanization for each province are considered when optimizing the emission pathways. Specifically, future urbanization rates and population are referred to the settings in refs. [50,51]. Economic structure is referred to the settings in refs. [49,52]. Please see Supplementary Tables 3, 4, and 5 for details.

### Reporting summary

Further information on research design is available in the Nature Portfolio Reporting Summary linked to this article.

## Data availability

All kinds of provincial energy consumption data were from the iNEMS database of the Center for Energy and Environmental Policy

Research of Beijing Institute of Technology[53]. National greenhouse gas inventory data were used for various energy emission factors, including 2.66 tons of $CO_2$/standard coal for coal, 1.73 tons of $CO_2$/standard coal for oil products, and 1.56 tons of $CO_2$/standard coal for natural gas. The historical GDP data for each province were from the National Bureau of Statistics, and the forecast data were from the high-speed and low-speed scenario data in the ref. 34 (See Supplementary Table 1 for specific settings). The GDP data were uniformly converted into GDP values with 2020 as the constant price through the GDP index. This study sets the upper and lower limits of GDP for each province in future years according to the deviation degree from the national GDP growth rate (See Supplementary Table 3). The historical population data for each province is from the National Bureau of Statistics[54], and the future population forecast is from ref. 50. The energy consumption per unit output value and the proportion of secondary industry in each province were derived from the National Bureau of Statistics and provincial statistical yearbooks. The historical and future urbanization rate data for each province are from the National Bureau of Statistics and ref. 48,51,52. Please see the indicators in Supplementary Data 1 file. The future settings of urbanization and secondary industrial share can be seen in Supplementary Tables 4 and 5. The planning goals for each province, such as the year of carbon peak, energy intensity, coal proportion, non-fossil energy proportion, and other data, were collected from the official policy documents launched by provincial governments (Supplementary Table 2).

## Code availability

Nonlinear regression was used to investigate the relationship between socio-economic development, technology, and energy. The corresponding equations are illustrated in the Methods section of the manuscript. The code for regression is written and calculated by IBM SPSS Statistics 26.0.0 and related descriptions of this software can be accessed at https://www.ibm.com/docs/en/spss-statistics/26.0.0. The nonlinear programming solution procedure was used to solve our Mr. COEP model with the equations illustrated in the Methods section of our manuscript. The nonlinear optimization solver is based on GAMS 24.8.3 and CONOPT Optimizer. The code and related description of GAMS Optimizer can be accessed at https://www.gams.com/latest/docs/RN_248.html.

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

## Acknowledgements

The authors gratefully acknowledge the financial support of National Natural Science Foundation of China (nos. 72225010 received by Biying Yu, 72293605 received by Yi-Ming Wei, 71822401 received by Biying Yu, 72073014 received by Lan-Cui Liu, 72104025 received by Jia-Ning Kang, 72140003 received by Biying Yu). We would like to thank dear Lele and our colleagues for their support and acknowledge help from CEEP-BIT.

## Author contributions

B.Y., Y.W. and L.L. conceived the study and performed the analysis. Z.Z. analyzed the data and implemented the model. Z.Z. and Q.Z. contributed to the data collection and processing. Z.Z. and S.X. contributed to preparing the figures. B.Y., J.K. and H.L. worked on the review and editing. All authors approved and contributed to writing the paper.

## Competing interests

The authors declare no competing interests.
