## [Peer Review File · Nature Communications]

Approaching national climate targets in China considering the challenge of regional inequalityREVIEWER COMMENTS

Reviewer #1 (Remarks to the Author):

Equitable and ambitious regional mitigation efforts are critical to supporting the national plan to achieve carbon neutrality goals. Current studies focus more on pathways consistent with carbon peaking and carbon neutrality at the national level, without scrutinizing the implications of equity and growth at the local level. This study combines an Integrated Assessment Model (IAM) with an optimization model based on Kuznets' theory to study the optimal allocation scheme among different provinces in China. Although the topic is pertinent, I have some concerns regarding the methodology adopted in this study.

Major Concerns:

1. The optimization model used in this study is based on an extended Kuznets framework calibrated with historical data at the provincial level. Could the authors justify how these historical trends stand for the long-term future until 2060, given the economic structure will be substantially different? In addition, how can this framework address the complicated economic linkage among different provinces? The shock on one province may have a significant impact on others as well. From my perspective, the Mr COEP framework adopted in this study may not adequately address these concerns.

2. This study highlights the development of an equity index as its key contribution. However, the methodology of such an equity index is not clear enough. In the method section, only a qualitative explanation is given without a clear and careful discussion on how different provinces are categorized into four groups and how those equity indices are taken as constraints. Additionally, this paper mixes the use of "maturity index" with "equity index", it would be better if the use of these terms were consistent.

Minor Concerns:

1. Regarding Figure 1, what is the pink ribbon in panel a? Is it the uncertainty range or something else? In addition, this figure includes ten scenarios, but the following discussion only focuses on the LD-HT scenario. Why do the authors present the other nine scenarios in this figure without discussion?

2. In lines 200-202, the authors state, "Specifically, compared with the AP30 scenario, 26 of 30 provinces will have additional GDP gains, ranging from 18 to 3981 billion RMB and 1.1 billion RMB on average during 2023–2040 if following the collaborative strategy." Why do 26 provinces have additional GDP gains ranging from 18 to 3981 billion RMB, but the average is much lower than this range, with only 1.1 billion RMB?

3. The results section spends too much space showing the numbers from various scenarios without a clear presentation of the findings.

4. Regarding Figure 6a, why is Beijing missing in the collaborative strategy?

5. When the authors use IAM to define carbon peaking and carbon neutrality, it is clear that they use the scope of Fossil Fuel and Industrial Emissions (FFI) and carbon sink. But when they move to the regional discussion, it seems they only focus on energy-related emissions (formula 6 in the method section). It is important to keep the emission scope consistent.

6. In line 83, the authors state "and peak CO₂ emissions will be approximately 12.2 billion tons (including industrial process emissions)." while in line 302, the peaking level is 12.7 billion tons, also with industrial process emissions included. Which peaking level is correct?

7. The authors should discuss the shortcomings of this study at the end of the conclusion, and key uncertainties which may impact the findings.

Reviewer #2 (Remarks to the Author):

The paper provides very interesting and unique perspectives related to achieving national climate targets and ensuring the greatest local and national benefits as well as regional equality. The findings have significant policy implications and the imperative of bottom up/regional planning is evident. In the context of China there has been a number of studies that have explored abatement pathway at the sectoral level, however regional level analysis is scarce. The paper is timely and it is carried out in a very comprehensive way. However, the authors are requested to go through the following comments and feedback and accordingly revise the paper.

Comments:

- (1) It is recommended that the detailed Settings for HD, MD, and LD, as well as HT, MT, and LT in Fig. 1 can be described in the supplementary Material. The current version is not very clear.
- (2) Please explain further which year the data is in Fig. 2.
- (3) The authors proposed a regional equity index, please further explain the principles and basis for constructing this index.
- (4) The authors should add much more recent literature on multi-regional analysis in China, for example Zhang et al. (2022; *Climate Policy*, <https://doi.org/10.1080/14693062.2022.2119198>); Jiang et al. (2023; *Ecological Economics*, <https://doi.org/10.1016/j.ecolecon.2022.107675>).
- (5) Some results are not deeply analyzed, please strengthen the economic analysis and interpretation of the main results, for example, Lines 75-97 show only the results without economic explanation.
- (6) In lines 160-164, the author states that the EIP30 scenario refers to some scholars and policy makers, and the author should provide the corresponding reference basis.
- (7) Detailed settings of changes in GDP, population, urbanization rate, etc. for each province under the baseline scenario should be added to the supplementary material. It is not enough that the author only introduces the reference basis for these variables.
- (8) There are some minor grammatical errors in your paper, please make a proofreading or double check about your whole paper.

Response to reviewers' comments

We sincerely appreciate the reviewer for his/her insightful review. The comments and suggestions have contributed substantially to improve our paper. We have tried our best to revise the manuscript. Our point-by-point responses are as follows. The revised portions extracted from the article are indicated in *italics*.

■ To Reviewer #1's comments:

Equitable and ambitious regional mitigation efforts are critical to supporting the national plan to achieve carbon neutrality goals. Current studies focus more on pathways consistent with carbon peaking and carbon neutrality at the national level, without scrutinizing the implications of equity and growth at the local level. This study combines an Integrated Assessment Model (IAM) with an optimization model based on Kuznets' theory to study the optimal allocation scheme among different provinces in China. Although the topic is pertinent, I have some concerns regarding the methodology adopted in this study.

Response :

Thanks to the reviewer for your positive comments. We tried our best to revise our work and addressed all comments as follows. Please note that since China's target year for achieving carbon neutrality is 2060, we extended the research time period from 2040 to 2060 in this revised manuscript. All the results have been updated.

Major Concerns:

1. The optimization model used in this study is based on an extended Kuznets framework calibrated with historical data at the provincial level. Could the authors justify how these historical trends stand for the long-term future until 2060, given the economic structure will be substantially different? In addition, how can this framework address the complicated economic linkage among different provinces? The shock on one province may have a significant impact on others as well. From my perspective,

the Mr COEP framework adopted in this study may not adequately address these concerns.

Response :

Thank you for your comment. We'd like to explain from three aspects:

(1) Regarding the concern about “how these historical trends stand for the long-term future until 2060”, yes, we used historical data to quantify the relationship between energy consumption, economic development level, and other important influencing factors in each province based on an extended Kuznets framework. However, to represent the future trends, we have considered the dynamic changes of main influencing factors (including secondary industry share, population, and urbanization) in future for each province when conducting the Mr. COEP model to optimize the emission pathways. Specifically, for future urbanization rates and population, we referred to the settings in references 36 and 37. For the economic structure, we referred to the settings in references 38 and 39. We have added the future dynamics of these factors to the supplementary documents (see Supplementary Table 3, Table 4, and Table 5). By incorporating the changes of these socio-economic factors into Mr. COEP model, we can to some extent capture the future trends of social and economic development for each region. We have clarified the above information in the “Methods-Nonlinear relationship between socio-economic development, technology, and energy” section as: *“To represent the future trends of social and economic development, the dynamic changes of secondary industry share, population, and urbanization for each province are considered when optimizing the emission pathways. Specifically, future urbanization rates and population are referred to the settings in references 36 and 37. Economic structure is referred to the settings in references 38 and 39. Please see Supplementary Table 3, Table 4, and Table 5 for details.”*

(2) In our Mr. COEP model, the economic linkage among different provinces is represented through the constraints of national energy consumption and carbon emissions as well as the objective function. Specifically, regional carbon emissions are optimized under the constraints of annual national emissions and energy consumption by fuel type (i.e., coal, oil, gas, non-fossil consumption) which are derived from C³IAM/NET model. Because regional GDP is linked with its energy consumption

following the nonlinear relationship (see Methods), thus, economic changes in one region will lead to changes in its energy consumption by fuel type, which will then change the energy consumption for other regions. This change will further influence the GDP of other regions and then the national GDP. The optimization will iterate the above process until the objective (maximize the national GDP) is satisfied. Finally, regional carbon emissions are calculated based on their fossil energy consumption. We have added the above explanation in the “Method” section.

(3) Though we have tried our best to consider the future dynamics of socio-economic factors and the linkage among different provinces in Mr. COEP model, there may be a large or even disruptive impact on aspects such as regional industrial structure due to the accelerated carbon neutrality process and uneven resource distribution. Therefore, we have mentioned this as a limitation of our study and highlighted that such uncertainties should be further investigated in the future study: *“the accelerated process of carbon neutrality may lead to disruptive industry transfer across provinces due to uneven distribution of low-carbon technologies and renewable resource, thereby changing the economic linkages among provinces. Though this study has taken into account the influence of dynamic socio-economic development in future following the literature prediction, further uncertainties could be investigated in the next step”*.

2. This study highlights the development of an equity index as its key contribution. However, the methodology of such an equity index is not clear enough. In the method section, only a qualitative explanation is given without a clear and careful discussion on how different provinces are categorized into four groups and how those equity indices are taken as constraints. Additionally, this paper mixes the use of "maturity index" with "equity index", it would be better if the use of these terms were consistent.

Response :

Thank you for your valuable comment. And sorry for the confusion of maturity index and equity index.

(1) Regarding the index name, we have unified the name to “maturity index”, which is used to indicate the equity on achieving the carbon mitigation targets for

different provinces. To avoid confusion and motivated by the reviewer’s valuable comment, in this revised manuscript, we use the TOPSIS (Technique for Order Preference by Similarity to an Ideal Solution) method to calculate the maturity index score for each province. The indicators used in TOPSIS analysis is the same with previous cluster analysis, including GDP, per capita GDP, energy intensity, proportion of coal consumption, and installed capacity of renewable energy for each province. Among them, GDP, per capita GDP, and renewable energy installed capacity are positive indicators, while energy intensity and the proportion of coal consumption are negative indicators. For provinces with lower maturity scores (e.g., economically backward or coal-dominant provinces or provinces with less renewable resource), it is more challenging and unfair for them to achieve the carbon mitigation targets at the same time with other provinces. The specific TOPSIS results are shown in the Fig.2b and Supplementary Fig. 2). An explanation has been added in the “Methods” section.

(2) Regarding how the maturity index is used as constraint in the Mr. COEP model, it works as follows: we first divided the provinces into three groups based on the maturity index score derived from the TOPSIS method, which are high-maturity provinces, middle-maturity provinces, and low-maturity provinces. Then we set the constraints of the carbon peak sequence for these three groups of provinces in the model under the Collaborative optimization (COP) scenario. Specifically, the carbon peak year for high-maturity provinces is constrained to be earlier than that of middle-maturity provinces; and carbon peak year for middle-maturity provinces is constrained to be earlier than that of low-maturity provinces. Then the exact carbon peak year for each province is optimized in the model by considering carbon peak sequence constraint as well as national and regional economic benefits. We have added the explanation in the figure caption of Fig. 2 and revised the descriptions in the Methods section as: *“Provinces are then divided into high-maturity, middle-maturity and low-maturity groups based on their maturity index score. Subsequently, we set the constraints of the carbon peak sequence for these three groups of provinces in the model under the COP scenario. Specifically, the carbon peak year for high-maturity provinces is constrained to be earlier than that of middle-maturity provinces; and carbon peak year for middle-maturity provinces is constrained to be earlier than that of low-maturity provinces.”*

Then the exact carbon peak year for each province is optimized by considering carbon peak sequence constraint as well as national and regional economic benefits”.

Minor Concerns:

1. Regarding Figure 1, what is the pink ribbon in panel a? Is it the uncertainty range or something else? In addition, this figure includes ten scenarios, but the following discussion only focuses on the LD-HT scenario. Why do the authors present the other nine scenarios in this figure without discussion?

Response :

Thank you for your comment and sorry for the confusion. Actually, we apply the C³IAM/NET model to investigate the carbon peak and carbon neutrality pathway for China by considering two types of uncertainties, including ① future product (or service) demand in energy-consuming sectors corresponding to different socio-economic development speeds, and ② the carbon sink potential which would determine the desired transition speed for energy system (fewer carbon sink in 2060, then a higher transition speed is required for the energy system, and vice versa). By integrating these two uncertainties, we designed ten scenarios for investigating the carbon peak and carbon neutral pathway for China, which are BAU, and a combination of three levels for the demand (HD, MD, and LD) and three levels for the carbon sink potential (HT, MT, and LT). The pink ribbon in Fig. 1a is the uncertainty range. While for the regional model analysis, we only used the carbon emission pathway corresponding to the scenario of medium demand related to medium GDP growth (MD) and the carbon sink being 1 billion tons (HT) for China in 2060 as the constraint. The reason is that ① according to the current planning of China, the medium GDP growth is more likely to occur, and ② according to the literature 28, 1 billion tons of carbon sink in 2060 is more possible for China. In this revised manuscript, to avoid confusion, we only show the results of BAU scenario and the MD-HT scenario in Fig. 1a in the main text. The name of HT is modified to LCS, which means low carbon sink potential. The other scenarios are present in Supplementary Fig. 1. The descriptions in the main text have been modified to only introduce the results of the MD-LCS scenario. We also explained the above reason in the figure caption of Supplementary Fig. 1 Thank you again for

your valuable comment.

2. In lines 200-202, the authors state, "Specifically, compared with the AP30 scenario, 26 of 30 provinces will have additional GDP gains, ranging from 18 to 3981 billion RMB and 1.1 billion RMB on average during 2023–2040 if following the collaborative strategy." Why do 26 provinces have additional GDP gains ranging from 18 to 3981 billion RMB, but the average is much lower than this range, with only 1.1 billion RMB?

Response :

We are very sorry for the unit error here. We have checked the GDP gains of each province and it should be "1.1 trillion". In the revised manuscript, the results have been updated and the unit is modified. Thank you for pointing out our mistake.

3. The results section spends too much space showing the numbers from various scenarios without a clear presentation of the findings.

Response :

Thank you for your valuable comment. In the result section, we try to delete some numbers and highlight the findings. For example, we added Fig. 4c to summarize the characteristics of carbon intensity change for different province groups. And the descriptions have also been revised. Please see the "Regional cost-effective pathways of carbon emissions and energy consumption" section. These quantitative results "*once again demonstrate that setting an early carbon peak target or having provinces operate separately is irrational; and this would cause national losses and harm the local economy for most provinces*". Based on these evidence, we proposed suggestions on how to revise the current planning on carbon peak time and energy structure for each province in the Discussion section.

Fig. 4 | Collaborative carbon mitigation strategy for each province. c. Five-year reduction rate of CO₂ emission intensity for provinces in three peak year groups. The icons indicate the values for each province during different time periods (2020-2025, 2025-2030, 2030-2035, and 2035-2040).

4. Regarding Figure 6a, why is Beijing missing in the collaborative strategy?

Response :

Thank you for your comment. Because the historical data shows that Beijing has already peaked its carbon emissions, so it was not shown in the figure in the previous manuscript. In the revised version, we have updated Fig. 6a to include Beijing. The modified figure is shown below.

Fig. 6 | Suggested settings of carbon mitigation and energy transition targets for provinces. a,

suggestions for the carbon peak year.

5. When the authors use IAM to define carbon peaking and carbon neutrality, it is clear that they use the scope of Fossil Fuel and Industrial Emissions (FFI) and carbon sink. But when they move to the regional discussion, it seems they only focus on energy-related emissions (formula 6 in the method section). It is important to keep the emission scope consistent.

Response :

Thanks a lot for your valuable comment. We apologize that we did not explain this clearly in the previous manuscript. When using the C³IAM/NET model to investigate China's carbon peak and carbon neutral pathway, our results include Fossil Fuel, Industrial Process Emissions and carbon sink. Therein, we set different scenarios based on the potential of carbon sink in 2060 and then optimize the technology portfolio for each sector in the energy system under the target of carbon neutrality. The carbon emissions obtained from C³IAM/NET model shown in Fig. 1a is the sum of fossil fuel-related emissions and industrial process emissions. In the previous manuscript, we use this total emission to conduct the regional analysis. But motivated by your valuable comment, we feel it may be not appropriate to include the industrial process emissions because it is more related to the distribution of the industries that will generate process emissions and cannot be explained by the provincial energy consumption. Considering that the fossil fuel-related emissions accounted for approximately 90% of total carbon emissions in China, so in the revised manuscript, we only use the fossil fuel related carbon emissions as the constraint in the regional analysis. We have updated all the results and clarified this in the Method section. And we also mentioned that “*the industrial process emissions across different regions need to be further investigated*” in the discussion part.

6. In line 83, the authors state "and peak CO₂ emissions will be approximately 12.2 billion tons (including industrial process emissions)." while in line 302, the peaking level is 12.7 billion tons, also with industrial process emissions included. Which

peaking level is correct?

Response :

Thank you very much for your valuable suggestion. “12.7 billion tons” is the maximum value under all scenarios. In this revised manuscript, we only introduced the results of one scenario in the entire text, which is 12.2 billion tons as mentioned earlier.

7. The authors should discuss the shortcomings of this study at the end of the conclusion, and key uncertainties which may impact the findings.

Response :

Thanks a lot for your suggestion. We have added the shortcomings of this paper and directions for future research in the last paragraph of the discussion section. The details are as follows.

“This research also has some limitations that can be improved in future study. For instance, only energy-related carbon emissions were considered here for provincial emission pathway optimization and the industrial process emissions across different regions need to be further investigated. In addition, the accelerated process of carbon neutrality may lead to disruptive industry transfer across provinces due to uneven distribution of low-carbon technologies and renewable resource, thereby changing the economic linkages among provinces. Though this study has taken into account the influence of dynamic socio-economic development in future following the literature prediction, further uncertainties should be explored in the next step”.

■ To Reviewer #2's comments:

The paper provides very interesting and unique perspectives related to achieving national climate targets and ensuring the greatest local and national benefits as well as regional equality. The findings have significant policy implications and the imperative of bottom up/regional planning is evident.

In the context of China there has been a number of studies that have explored abatement pathway at the sectoral level, however regional level analysis is scarce. The paper is timely and it is carried out in a very comprehensive way. However, the authors are requested to go through the following comments and feedback and accordingly revise the paper.

Response :

Thanks to the reviewer for your positive comments. We tried our best to revise our work and addressed all comments as follows. Please note that since China's target year for achieving carbon neutrality is 2060, we extended the research time period from 2040 to 2060 in this revised manuscript. And all the results have been updated.

Comments:

(1) It is recommended that the detailed Settings for HD, MD, and LD, as well as HT, MT, and LT in Fig. 1 can be described in the supplementary Material. The current version is not very clear.

Response :

Thank you for your suggestion. HD, MD, and LD indicate high, middle, low demand for product (or service) in energy-consuming sectors corresponding to different socio-economic development speeds. HT, MT, and LT indicate high-speed, mid-speed, and low-speed promotion for the low-carbon technologies or clean fuels in the energy system caused by low, middle, and high carbon sink potential. In other words, low carbon sink potential would require a high-speed transition for the energy system. In the revised version, we have changed the name of HT, MT, and LT to LCS, MCS, and HCS (CS means carbon sink), respectively. Actually, we apply the C³IAM/NET model to investigate the carbon peak and carbon neutrality pathway for China by considering

two types of uncertainties, including ① future product (or service) demand in energy-consuming sectors corresponding to different socio-economic development speeds, and ② the carbon sink potential which would determine the desired transition speed for energy system. By integrating these two uncertainties, we designed ten scenarios for investigating the carbon peak and carbon neutrality pathway for China, including BAU, and a combination of three levels for the demand (HD, MD, and LD) and three levels for the carbon sink potential (HCS, MCS, and LCS). The detailed setting related to demand and carbon sink potential for different scenarios is given in the Supplementary Document (see Supplementary Table 1 and Supplementary Fig. 1).

(2) Please explain further which year the data is in Fig. 2.

Response :

Thank you for your suggestion. We have added the data year used in the annotation of Figure 2 as: “*All the data is for year 2020.*”

(3) The authors proposed a regional equity index, please further explain the principles and basis for constructing this index.

Response :

Thank you for your valuable comment. First we apologize that there is some confusion between maturity index and equity index. We have unified the index name to be “maturity index”, which is used to quantify the capability and potential of each province to achieve the carbon peak and carbon neutral target. In this revised manuscript, we use the TOPSIS (Technique for Order Preference by Similarity to an Ideal Solution) method to calculate the maturity index score for each province. The indicators used in TOPSIS analysis is the same with previous cluster analysis, including GDP, per capita GDP, energy intensity, proportion of coal consumption, and installed capacity of renewable energy for each province. Among them, GDP, per capita GDP, and renewable energy installed capacity are positive indicators, while energy intensity and the proportion of coal consumption are negative indicators. For provinces with

lower maturity scores (e.g., economically backward or coal consuming provinces or provinces with less renewable resource), it is more challenging and unfair for them to achieve the carbon mitigation targets at the same time with other provinces. The specific TOPSIS results are shown in the Fig.2b and Supplementary Fig. 2). Based on the maturity index score, we grouped the provinces into three categories and all the results have been updated. Detailed explanation has been added to the “Methods” section as: *“Regional carbon peak maturity constraint. To reflect the differences and ensure equity among regions, this study innovatively proposes the regional maturity index and establishes an evaluation system with multiple indicators, including per capita GDP, energy intensity, coal consumption proportion, and installed renewable energy capacity, thus reflecting the level of regional economic development, energy and industrial structures, and technical level. Provinces are then divided into high-maturity, middle-maturity and low-maturity groups based on their maturity index score. Subsequently, we set the constraints of the carbon peak sequence for these three groups of provinces in the model under the COP scenario ...The TOPSIS (Technique for Order Preference by Similarity to an Ideal Solution) method is adopted to evaluate the maturity index of each province”*.

(4) The authors should add much more recent literature on multi-regional analysis in China, for example Zhang et al. (2022; Climate Policy, <https://doi.org/10.1080/14693062.2022.2119198>); Jiang et al. (2023; Ecological Economics, <https://doi.org/10.1016/j.ecolecon.2022.107675>).

Response :

Thanks a lot for your suggestion. We have added these references as: *“Prior literature has proposed a series of climate mitigation strategies at the global or national scale^{14,16–20”}*.

(5) Some results are not deeply analyzed, please strengthen the economic analysis and interpretation of the main results, for example, Lines 75-97 show only the results

without economic explanation.

Response :

Thanks a lot for your valuable suggestion. We optimize the entire energy system by minimizing total costs. We have considered the production and consumption costs of multiple sectors, including primary energy supply, electricity, heating, steel, cement, non-ferrous metals, chemical, residential, transportation, and other industries. The result obtained is a technology portfolio that can achieve carbon peak and carbon neutrality goals at the lowest cost. We have introduced the economic meaning in the Methods section.

(6) In lines 160-164, the author states that the EIP30 scenario refers to some scholars and policy makers, and the author should provide the corresponding reference basis.

Response :

Thanks a lot for your valuable suggestion. We have added the basis for setting up the EIP30 scenario, as: *“The third scenario is that Only energy-intensive provinces are required to achieve carbon peak before 2030 (EIP30). The main energy consuming provinces in China (including Hebei, Shanxi, Inner Mongolia, Liaoning, Jiangsu, Zhejiang, Anhui, Shandong, Henan, Guangdong, Shaanxi, and Xinjiang) accounted for approximately 70% of the country’s total carbon emissions in 2020. It is argued that if these provinces can reach their carbon emissions peak on time, then the entire country can achieve the target on time. Hence, we set up the EIP30 scenario”*.

(7) Detailed settings of changes in GDP, population, urbanization rate, etc. for each province under the baseline scenario should be added to the supplementary material. It is not enough that the author only introduces the reference basis for these variables.

Response :

Thank you for your suggestion. We have added the settings for each parameter in the Supplementary document. Please see Supplementary Table 3, Table 4, and Table 5.

(8) There are some minor grammatical errors in your paper, please make a proofreading or double check about your whole paper.

Response :

Thanks a lot for your suggestion. Our manuscript has been proofread by English native company and we also tried our best to improve the language presentation.

REVIEWER COMMENTS

Reviewer #1 (Remarks to the Author):

While the authors have made efforts to address some of my concerns, there are still significant issues that need to be resolved before this paper can be considered for publication.

1. Regarding the Mr. COEP optimization model presented in the article, it is theoretically sound that a reduction in constraints or the relaxation of constraints in the model would lead to a larger objective function, in this case, the sum of GDP of each province as discussed in the article. Since the primary focus of this study is to investigate how China can reduce the impact of carbon peaks on total GDP through cooperation among provinces, it raises a valid question as to why the authors did not consider a scenario in which constraints on peak times for provinces are not predefined, allowing the model optimization to determine the optimal peak times for each province. Theoretically, such an approach could yield better results compared to the scenarios predefined by the authors. It appears that such a peaking scenario may not be inherently policy irrelevant, and it would be beneficial for the authors to address this consideration in their discussion.

2. While the article outlines the objective function and various constraints of the Mr. COEP model within the methodology section, there is a lack of clarity regarding which parameters of the model are exogenously defined and which variables are endogenously determined by the model. This distinction is crucial for readers to grasp the mechanism and architecture of the model. It is strongly recommended that the authors provide this essential information for the benefit of their readers' understanding.

3. A pivotal aspect of Mr. COEP's model lies in its econometric model based on extended Kuznets curves. Although the authors have included a table of parameter estimates for this model in the supplementary information (SI) and provided information about the software package used in the Code availability section, there is a notable absence of comprehensive details concerning the data utilized for this econometric model. Specifically, information about the source of the model data and data tables is not adequately provided. In the interests of transparency and reproducibility, it is advisable that the authors disclose the data underpinning their econometric model, including the data source and relevant tables. Such practices align with the increasing standards of transparency seen in a growing number of economics journals and contribute to the rigor and reliability of the research.

Reviewer #2 (Remarks to the Author):

Thank you for addressing my comments. I find the revisions satisfactory and recommend publication.

Response to reviewers' comments

We sincerely appreciate the reviewer for his/her insightful review. The comments and suggestions have contributed substantially to improve our paper. We have tried our best to revise the manuscript. Our point-by-point responses are as follows. The revised portions extracted from the article are indicated in *italics* in this response letter, and they are also highlighted in yellow in the manuscript.

■ To Reviewer #1's comments:

While the authors have made efforts to address some of my concerns, there are still significant issues that need to be resolved before this paper can be considered for publication.

Response :

Thanks to the reviewer for your positive comments. We tried our best to revise our work and addressed all comments as follows.

1. Regarding the Mr. COEP optimization model presented in the article, it is theoretically sound that a reduction in constraints or the relaxation of constraints in the model would lead to a larger objective function, in this case, the sum of GDP of each province as discussed in the article. Since the primary focus of this study is to investigate how China can reduce the impact of carbon peaks on total GDP through cooperation among provinces, it raises a valid question as to why the authors did not consider a scenario in which constraints on peak times for provinces are not predefined, allowing the model optimization to determine the optimal peak times for each province. Theoretically, such an approach could yield better results compared to the scenarios predefined by the authors. It appears that such a peaking scenario may not be inherently policy irrelevant, and it would be beneficial for the authors to address this consideration in their discussion.

Response :

Thank you for your comment. We have estimated a scenario of not setting carbon peak time constraints for each province following your valuable suggestion. And we added the descriptions in the Discussion part as “*In addition to the above strategies, a scenario without any carbon peak time constraints is analyzed (see Supplementary Fig. 4-5 for details). The results show that though the total economic benefits would increase compared to that of COP scenario, some provinces with less-developed economy and fewer renewable resources (e.g., Shanxi and Hebei), which indicate lower carbon peak maturity, need to peak their carbon emissions earlier than that of COP, while some provinces with developed economy and rich renewable energy (e.g., Sichuan and Hubei) could delay their carbon peak year. This may be less feasible in practice. Therefore, it is once again highlighted that the carbon peak maturity that can represent the regional inequality needs to be considered during the decision making for the provincial carbon mitigation strategies*”. The reason behind this is that under the COP scenario, we considered the impact of economic level, industrial structure, renewable energy resource endowment, and energy structure when evaluating the provincial maturity of carbon peak, and the maturity category is incorporated into the Mr. COEP model so as to represent the regional inequality. In other words, **the results under COP scenario could better balance the economic benefits and regional equity, while the scenario of not setting carbon peak time constraints may be more focus on improving the economic benefit.**

Supplementary Fig. 4| The carbon peak years of provinces under the scenario of not setting carbon peak year constraints and COP scenario. Note that only provinces with different carbon peak years under two scenarios are shown here.

Supplementary Fig. 5| The economic benefits of each province under the scenario of not setting carbon peak year constraints compared with the results of COP scenario.

2. While the article outlines the objective function and various constraints of the Mr. COEP model within the methodology section, there is a lack of clarity regarding which parameters of the model are exogenously defined and which variables are endogenously determined by the model. This distinction is crucial for readers to grasp the mechanism and architecture of the model. It is strongly recommended that the authors provide this essential information for the benefit of their readers' understanding.

Response :

Thank you for your valuable comment. For the ease of understanding, we have clarified the definitions of each parameter in the model description section. Please see the “*In the Mr. COEP model, decision variables include $Energy_{i,k,t}$ (the consumption of coal, oil, gas, and non-fossil fuels in region i at year t) and $GDP_{i,t}$ (GDP of region i in year t). The remaining variables are exogenous variables, such as the national total amount of energy consumption by fuel type (which is derived from C^3IAM/NET model), population size, urbanization rate, industrial structure, etc. The specific settings of exogenous variables are shown in the Supplementary Tables 1-5.*”

3. A pivotal aspect of Mr. COEP's model lies in its econometric model based on extended Kuznets curves. Although the authors have included a table of parameter estimates for this model in the supplementary information (SI) and provided information about the software package used in the Code availability section, there is a notable absence of comprehensive details concerning the data utilized for this econometric model. Specifically, information about the source of the model data and data tables is not adequately provided. In the interests of transparency and reproducibility, it is advisable that the authors disclose the data underpinning their econometric model, including the data source and relevant tables. Such practices align with the increasing standards of transparency seen in a growing number of economics journals and contribute to the rigor and reliability of the research.

Response :

Thank you for your valuable comment. We have provided the data of the econometric model in Data file and also in the Supplementary documents.

■ To Reviewer #2's comments:

Thank you for addressing my comments. I find the revisions satisfactory and recommend publication.

Response :

We are appreciative of the reviewer's efforts and the comments, which helped us improve our research.

REVIEWERS' COMMENTS

Reviewer #1 (Remarks to the Author):

In response to the revisions made, the manuscript addresses my previous concerns and increases the transparency of methodology and data used in the analysis. Therefore, I suggest the proceeding of publication.